# Cornering Stationary and Restless Mixing Bandits
## with `Remix-UCB`

**Julien Audiffren**
CMLA
ENS Cachan, Paris Saclay University
94235 Cachan France
audiffren@cmla.ens-cachan.fr

**Liva Ralaivola**
QARMA, LIF, CNRS
Aix Marseille University
F-13289 Marseille cedex 9, France
liva.ralaivola@lif.univ-mrs.fr

## Abstract

We study the restless bandit problem where arms are associated with stationary $\varphi$-mixing processes and where rewards are therefore dependent: the question that arises from this setting is that of carefully recovering some independence by 'ignoring' the values of some rewards. As we shall see, the bandit problem we tackle requires us to address the exploration/exploitation/independence trade-off, which we do by considering the idea of a *waiting arm* in the new `Remix-UCB` algorithm, a generalization of `Improved-UCB` for the problem at hand, that we introduce. We provide a regret analysis for this bandit strategy; two noticeable features of `Remix-UCB` are that i) it reduces to the regular `Improved-UCB` when the $\varphi$-mixing coefficients are all 0, i.e. when the i.i.d scenario is recovered, and ii) when $\varphi(n) = O(n^{-\alpha})$, it is able to ensure a controlled regret of order $\widetilde{\Theta}\left(\Delta_*^{(\alpha-2)/\alpha} \log^{1/\alpha} T\right)$, where $\Delta_*$ encodes the distance between the best arm and the best suboptimal arm, even in the case when $\alpha < 1$, i.e. the case when the $\varphi$-mixing coefficients *are not* summable.

## 1   Introduction

**Bandit with mixing arms.** The bandit problem consists in an agent who has to choose at each step between $K$ arms. A stochastic process is associated to each arm, and pulling an arm produces a reward which is the realization of the corresponding stochastic process. The objective of the agent is to maximize its long term reward. In the abundant bandit literature, it is often assumed that the stochastic process associated to each arm is a sequence of independently and identically distributed (i.i.d) random variables (see, e.g. [12]). In that case, the challenge the agent has to address is the well-known exploration/exploitation problem: she has to simultaneously make sure that she collects information from all arms to try to identify the most rewarding ones—this is *exploration*—and to maximize the rewards along the sequence of pulls she performs—this is *exploitation*. Many algorithms have been proposed to solve this trade-off between exploration and exploitation [2, 3, 6, 12]. We propose to go a step further than the i.i.d setting and to work in the situation where the process associated with each arm is a stationary $\varphi$-mixing process: the rewards are thus dependent from one another, with a strength of dependence that weakens over time. From an application point of view, this is a reasonable dependence structure: if a user clicks on some ad (a typical use of bandit algorithms) at some point in time, it is very likely that her choice will have an influence on what she will click in the close future, while it may have a (lot) weaker impact on what ad she will choose to view in a more distant future. As it shall appear in the sequel, working with such dependent observations poses the question of how informative are some of the rewards with respect to the value of an arm since, because of the dependencies and the strong correlation between close-by (in time) rewards, they might not reflect the true 'value' of the arms. However, as the dependencies weaken over time, some kind of independence might be recovered if some rewards are ignored, in some sense. This

actually requires us to deal with a new tradeoff, the exploration/exploitation/independence tradeoff, where the usual exploration/exploitation compromise has to be balanced with the need for some independence. Dealing with this new tradeoff is the pivotal feature of our work.

**Non i.i.d bandit.** A closely related setup that addresses the bandit problem with dependent rewards is when they are distributed according to Markov processes, such as Markov chains and Markov decision process (MDP) [16, 22], where the dependences between rewards are of bounded range, which is what distinguishes those works with ours. Contributions in this area study two settings: the rested case, where the process attached to an arm evolves only when the arm is pulled, and the *restless* case, where all processes simultaneously evolve at each time step. In the present work, we will focus on the restless setting. The adversarial bandit setup (see e.g. [1, 4, 19]) can be seen as a non i.i.d setup as the rewards chosen by the adversary might depend on the agent's past actions. However, even if the algorithms developed for this framework can be used in our setting, they might perform very poorly as they are not designed to take advantage of any mixing structure. Finally, we may also mention the bandit scenario where the dependencies are between the arms instead being within-arm time-dependent (e.g., [17]); this is orthogonal to what we propose to study here.

**Mixing Processes.** Mixing process theory is hardly new. One of the seminal works on the study of mixing processes was done by Bernstein [5] who introduced the well-known block method, central to prove results on mixing processes. In statistical machine learning, one of the first papers on estimators for mixing processes is [23]. More recent works include the contributions of Mohri and Rostamizadeh [14, 15], which address the problem of stability bound and Rademacher stability for $\varphi$- and $\beta$-mixing processes; Kulkarni et al [11] establish the consistency of regularized boosting algorithms learning from $\beta$-mixing processes, Steinwart et al [21] prove the consistency of support vector machines learning from $\alpha$-mixing processes and Steinwart and Christmann [20] establish a general oracle inequality for generic regularized learning algorithms and $\alpha$-mixing observations. As far as we know, it is the first time that mixing processes are studied in a multi-arm bandit framework.

**Contribution.** Our main result states that a strategy based on the improved Upper Confidence Bound (or `Improved-UCB`, in the sequel) proposed by Auer and Ortner [2], allows us to achieve a controlled regret in the restless mixing scenario. Namely, our algorithm, `Remix-UCB` (which stands for Restless Mixing UCB), achieves a regret of the form $\widetilde{\Theta}(\Delta_*^{(\alpha-2)/\alpha} \log^{1/\alpha} T)$, where $\Delta_*$ encodes the distance between the best arm and the best suboptimal arm, $\alpha$ encodes the rate of decrease of the $\varphi$ coefficients, i.e. $\varphi(n) = O(n^\alpha)$, and $\widetilde{\Theta}$ is a $\mathcal{O}$-like notation (that neglects logarithmic dependencies, see Section 2.2). It is worth noticing that all the results we give hold for $\alpha < 1$, i.e. when the dependencies are *no longer summable*. When the mixing coefficients at hand are all zero, i.e. in the i.i.d case, the regret of our algorithm naturally reduces to the classical `Improved-UCB`. `Remix-UCB` uses the assumption about known (convergence rates of) $\varphi$-mixing coefficients, which is a classical standpoint that has been used by most of the papers studying the behavior of machine learning algorithms in the case of mixing processes (see e.g. [9, 14, 15, 18, 21, 23]). The estimation of the mixing coefficients poses a learning problem on its own (see e.g. [13] for the estimation of $\beta$-mixing coefficients) and is beyond the scope of this paper.

**Structure of the paper.** Section 2 defines our setup: $\varphi$-mixing processes are recalled, together with a relevant concentration inequality for such processes [10, 15]; the notion of regret we focus on is given. Section 3 is devoted to the presentation of our algorithm, `Remix-UCB`, and to the statement of our main result regarding its regret. Finally, Section 4 discusses the obtained results.

## 2  Overview of the Problem

### 2.1  Concentration of Stationary $\varphi$-mixing Processes

Let $(\Omega, \mathcal{F}, \mathbb{P})$ be a probability space. We recall the notions of stationarity and $\varphi$-mixing processes.

**Definition 1** (Stationarity)**.** *A sequence of random variables $\underline{X} = \{X_t\}_{t \in \mathbb{Z}}$ is* stationary *if, for any $t, m \geq 0$, $s \geq 0$, $(X_t, \ldots, X_{t+m})$ and $(X_{t+s}, \ldots, X_{t+m+s})$ are identically distributed.*

**Definition 2** ($\varphi$-mixing process)**.** *Let $\underline{X} = \{X_t\}_{t \in \mathbb{Z}}$ be a stationary sequence of random variables. For any $i, j \in \mathbb{Z} \cup \{-\infty, +\infty\}$, let $\sigma_i^j$ denote the $\sigma$-algebra generated by $\{X_t : i \leq t \leq j\}$. Then,*

*for any positive $n$, the $\varphi$-mixing coefficient $\varphi(n)$ of the stochastic process $\underline{X}$ is defined as*

$$\varphi(n) = \sup_{t, A \in \sigma_{t+n}^{+\infty}, B \in \sigma_{-\infty}^{t}, \mathbb{P}(B)>0} \left| \mathbb{P}\left[A|B\right] - \mathbb{P}\left[A\right] \right|. \tag{1}$$

$\underline{X}$ *is $\varphi$-mixing if $\varphi(n) \to 0$. $\underline{X}$ is algebraically mixing if $\exists \varphi_0 > 0, \alpha > 0$ so that $\varphi(n) = \varphi_0 n^{-\alpha}$.*

As we recall later, concentration inequalities are the pivotal tools to devise multi-armed bandits strategy. Hoeffding's inequality [7, 8] is, for instance, at the root of a number of UCB-based methods. This inequality is yet devoted to characterize the deviation of the sum of *independent* variables from its expected value and cannot be used in the framework we are investigating. In the case of stationary $\varphi$-mixing distributions, there however is the following concentration inequality, due to [10] and [15].

**Theorem 1** ([10, 15]). *Let $\psi_m : \mathcal{U}^m \to \mathbb{R}$ be a function defined over a countable space $\mathcal{U}$, and $\underline{X}$ be a stationary $\varphi$-mixing process. If $\psi_m$ is $\ell$-Lipschitz wrt the Hamming metric for some $\ell > 0$, then*

$$\forall \varepsilon > 0, \ \mathbb{P}_{\underline{X}}\left[|\psi_m(\underline{X}) - \mathbb{E}\psi_m(\underline{X})| > \varepsilon\right] \leq 2 \exp\left[-\frac{\varepsilon^2}{2m\ell^2\Lambda_m^2}\right], \tag{2}$$

*where $\Lambda_m \doteq 1 + 2\sum_{\tau=1}^{m} \varphi(\tau)$ and $\psi_m(\underline{X}) = \psi_m(\underline{X_0}, \dots, \underline{X_m})$.*

Here, we do not have to use this concentration inequality in its full generality as we will restrict to the situation where $\psi_m$ is the mean of its arguments, i.e. $\psi_m(X_{t_1}, \dots, X_{t_m}) \doteq \frac{1}{m}\sum_{i=1}^{m} X_{t_i}$, which is obviously $1/m$-Lipschitz provided that the $X_t$'s have range $[0; 1]$—which will be one of our working assumptions. If, with a slight abuse of notation, $\Lambda_m$ is now used to denote

$$\Lambda_m(\boldsymbol{t}) \doteq 1 + 2\sum_{i=2}^{m} \varphi(t_i - t_1), \tag{3}$$

for an increasing sequence $\boldsymbol{t} = (t_i)_{i=1}^{m}$ of times steps, then, the concentration inequality that will serve our purpose is given in the next corollary.

**Corollary 1** ([10, 15]). *Let $\underline{X}$ be a stationary $\varphi$ mixing process. The following holds: for all $\varepsilon > 0$ and all $m$-sequence $\boldsymbol{t} = (t_i)_{i=1}^{m}$ with $t_1 < \dots < t_m$,*

$$\mathbb{P}_{\{X_t\}_{t \in \boldsymbol{t}}}\left[\left|\frac{1}{m}\sum_{i=1}^{m} X_{t_i} - \mathbb{E}X_1\right| > \varepsilon\right] \leq 2\exp\left[-\frac{m\varepsilon^2}{2\Lambda_m^2(\boldsymbol{t})}\right]. \tag{4}$$

(Thanks to the stationarity of $\{X_t\}_{t \in \mathbb{Z}}$ and the linearity of the expectation, $\mathbb{E}\sum_{i=1}^{m} X_{t_i} = m\mathbb{E}X_{t_1}$.)

**Remark 3.** *According to Kontorovitch's paper [10], the function $\Lambda_m$ should be $\max_j \left\{1 + 2\sum_{i=j+1}^{m} \varphi(t_i - t_j)\right\}$. However, when the time lag between two consecutive time steps $t_i$ and $t_{i+1}$ is non-decreasing, which will be imposed by the* Remix-UCB *algorithm (see below), and the mixing coefficients are decreasing, which is a natural assumption that simply says that the amount of dependence between $X_t$ and $X_{t'}$ reduces when $|t - t'|$ increases, then $\Lambda_m$ reduces to the more compact expression given by* (3).

Note that when there is independence, then $\varphi(\tau) = 0$, for all $\tau$, $\Lambda_m = 1$ and, as a consequence, Equation (4) reduces to Hoeffding's inequality: the precise values of the time instants in $\boldsymbol{t}$ do not impact the value of the bound and the length $m$ of $\boldsymbol{t}$ is the central parameter that matters. This is in clear contrast with what happens in the dependent setting, where the bound on the deviation of $\sum_{i=1}^{m} X_{t_i}/m$ from its expectation directly depends on the timepoints $t_i$ through $\Lambda_m$. For two sequences $\boldsymbol{t} = (t_i)_{i=1}^{m}$ and $\boldsymbol{t'} = (t_i')_{i=1}^{m}$ of $m$ timepoints, $\sum_{i=1}^{m} X_{t_i}/m$ may be more sharply concentrated around $\mathbb{E}X_1$ than $\sum_{i=1}^{m} X_{t_i'}/m$ provided $\Lambda_m(\boldsymbol{t}) < \Lambda_m(\boldsymbol{t'})$, which can be a consequence of a more favorable spacing of the points in $\boldsymbol{t}$ than in $\boldsymbol{t'}$.

## 2.2 Problem: Minimize the Expected Regret

We may now define the multi-armed bandit problem we consider and the regret we want to control.

**Restless $\varphi$-mixing Bandits.** We study the problem of sampling from a $K$-armed $\varphi$-mixing bandit. In our setting, pulling arm $k$ at time $t$ provides the agent with a realization of the random variable $X_t^k$,

where the family $\left\{X_t^k\right\}_{t\in\mathbb{Z}}$ satisfies the following assumptions: (A) $\forall k$, $(X_t^k)_{t\in\mathbb{Z}}$ is a stationary $\varphi$-mixing process with decreasing mixing coefficients $\varphi_k$ and (B) $\forall k$, $X_1^k$ takes its values in a discrete finite set (by stationarity, the same holds for any $X_t^k$, with $t\neq 1$) included in $[0;1]$.

**Regret** The regret we want to bound is the classical pseudo-regret which, after $T$ pulls, is given by

$$\mathcal{R}(T) \doteq T\mu^* - \mathbb{E}\sum_{t=1}^{T}\mu_{I_t} \tag{5}$$

where $\mu_k \doteq \mathbb{E}X_1^k$, $\mu^* \doteq \max_k \mu_k$, and $I_t$ is the index of the arm selected at time $t$. We want to devise a strategy is capable to select, at each time $t$, the arm $I_t$ so that the obtained regret is minimal.

**Bottleneck.** The setting we assume entails the possibility of long-term dependencies between the rewards output by the arms. Hence, as evoked earlier, in order to choose which arm to pull, the agent is forced to address the exploration/exploitation/independence trade-off where *independence* may be partially recovered by taking advantage of the observation regarding spacings of timepoints that induce sharper concentration of the empirical rewards than others. As emphasized later, targetting good spacing in the bandit framework translates into the idea of ignoring the rewards provided by some pulls to compute the empirical averages: this idea is carried by the concept of a *waiting arm*, which is formally defined later on. The questions raised by the waiting arm that we address with the `Remix-UCB` algorithm are a) how often should the waiting arm be pulled so the concentration of the empirical means is high enough to be relied on (so the usual exploration/exploitation tradeoff can be tackled) and b) from the regret standpoint, how hindering is it to pull the waiting arm?

$\mathcal{O}$ **and $\widetilde{\Theta}$ analysis.** In the analysis of `Remix-UCB` that we provide, just as is the case for most, if not all, analyses that exist for bandit algorithms, we will focus in the order of the regret and we will not be concerned about the precise constants involved in the derived results. We will therefore naturally heavily rely on the usual $\mathcal{O}$ notation and on the $\widetilde{\Theta}$ notation, that bears the following meaning.

**Definition 4** ($\widetilde{\Theta}$ notation). *For any two functions $f,g$ from $\mathbb{R}$ to $\mathbb{R}$, we say that $f = \widetilde{\Theta}(g)$ if there exist $\alpha,\beta > 0$ so that $|f|\log^{\alpha}|f| \leq |g|$, and $|g|\log^{\beta}|g| \leq |f|$.*

## 3 `Remix-UCB`: a UCB Strategy for Restless Mixing Bandits

This section contains our main contributions: the `Remix-UCB` algorithm. From now on, we use $a \vee b$ (resp. $a \wedge b$) for the maximum (resp. minimum) of two elements $a$ and $b$. We consider that the processes attached to the arms are algebraically mixing and for arm $k$, the exponent is $\alpha_k > 0$: there exist $\varphi_{k,0}$ such that $\varphi_k(t) = \varphi_{k,0}t^{-\alpha_k}$—this assumption is not very restrictive as considering rates such as $t^{-\alpha_k}$ are appropriate/natural to capture and characterize the decreasing behavior of the convergent sequence $(\varphi_k(t))_t$. Also, we will sometimes say that arm $k$ is faster (resp. slower) than arm $k'$ for $k \neq k'$, to convey the fact that $\alpha_k > \alpha_{k'}$ (resp. $\alpha_k < \alpha'_k$).

For any $k$ and any increasing sequence $\boldsymbol{\tau} = (\tau(n))_{n=1}^{t}$ of $t$ timepoints, the empirical reward $\widehat{\mu}_k^{\boldsymbol{\tau}}$ of $k$ given $\boldsymbol{\tau}$ is $\widehat{\mu}_k^{\boldsymbol{\tau}} \doteq \frac{1}{t}\sum_{n=1}^{t}X_{\tau(n)}^k$. The subscripted notation $\boldsymbol{\tau}_k = (\tau_k(n))_{1\leq n\leq t}$ is used to denote the sequence of timepoints at which arm $k$ was selected. Finally, we define $\Lambda_k^{\boldsymbol{\tau}}$ in a similar way as in (3), the difference with the former notation being the subscript $k$, as

$$\Lambda_k^{\boldsymbol{\tau}_k} \doteq 1 + 2\sum_{n=1}^{t}\varphi_k(\tau_k(n) - \tau_k(1)). \tag{6}$$

We feel important to discuss when `Improved-UCB` may be robust to the mixing process scenario.

### 3.1 Robustness of `Improved-UCB` to Restless $\varphi$-Mixing Bandits

We will not recall the `Improved-UCB` algorithm [2] in its entirety as it will turn out to be a special case of our `Remix-UCB` algorithm, but it is instructive to identify its distinctive features that make it a relevant base algorithm for the handling of mixing processes. First, it is essential to keep in mind that `Improved-UCB` is designed for the i.i.d case and that it achieves an optimal $\mathcal{O}(\log T)$ regret. Second, it is an algorithm that works in successive rounds/epochs, at the end of each of which a number of arms are eliminated because they are identified (with high probability) as being the least

promising ones, from a regret point of view. More precisely, at each round, the same number of *consecutive* pulls is planned for each arm: this number is induced by Hoeffding's inequality [8] and devised in such a way that all remaining arms share the same confidence interval for their respective expected gains, the $\mu_k = \mathbb{E}X_1^k$, for $k$ in the set of remaining arms at the current round. From a technical standpoint, this is what makes it possible to draw conclusions on whether an arm is useless (i.e. eliminated) or not. It is enlightening to understand what are the favorable and unfavorable setups for `Improved-UCB` to keep working when facing restless mixing bandits. The following Proposition depicts the favorable case.

**Proposition 5.** *If $\sum_t \varphi_k(t) < +\infty$, $\forall k$, then the classical* `Improved-UCB` *run on the restless $\varphi$-mixing bandit preserves its $\mathcal{O}(\log T)$ regret.*

*Proof.* Straightforward. Given the assumption on the mixing coefficients, it exists $M > 0$ such that $\max_{k \in \{1, \cdots, K\}} \sum_{t \geq 0} \varphi_k(t) < M$. Therefore, from Theorem 1, for any arm $k$, and any sequence $\boldsymbol{\tau}$ of $|\boldsymbol{\tau}|$ consecutive timepoints, $\mathbb{P}\left(|\mu_k - \widehat{\mu}_k^{\boldsymbol{\tau}}| > \varepsilon\right) \leq 2 \exp\left(-\frac{|\boldsymbol{\tau}|\varepsilon^2}{2(1+2M)^2}\right)$, which is akin to Hoeffding's inequality up to the multiplicative $(1 + 2M)^2$ constant in the exponential. This, and the lines to prove the $\mathcal{O}(\log T)$ regret of `Improved-UCB` [2] directly give the desired result. $\square$

In the general case where $\sum_t \varphi_k(t) < +\infty$ does not hold for every $k$, then nothing ensures for `Improved-UCB` to keep working, the idea of consecutive pulls being the essential culprit. To illustrate the problem, suppose that $\forall k$, $\varphi_k(n) = n^{-1/4}$. Then, after a sequence $\boldsymbol{\tau} = (t_1 + 1, t_1 + 2, \ldots, t_1 + t)$ of $t$ consecutive time instances where $k$ was selected, simple calculations give that $\Lambda_k^{\boldsymbol{\tau}} = \mathcal{O}(t^{3/4})$ and the concentration inequality from Corollary 1 for $\widehat{\mu}_k^{\boldsymbol{\tau}}$ reads as

$$\mathbb{P}(|\mu_k - \widehat{\mu}_k^{\boldsymbol{\tau}}| > \varepsilon) \leq 2 \exp\left(-C\varepsilon^2 t^{-1/2}\right) \tag{7}$$

where $C$ is some strictly positive constant. The quality of the confidence interval that can be derived from this concentration inequality *degrades* when additional pulls are performed, which counters the usual nature of concentration inequalities and prevents the obtention of a reasonable regret for `Improved-UCB`. This is a direct consequence of the dependency of the $\varphi$-mixing variables. Indeed, if $\varphi(n)$ decreases slowly, taking the average over multiple consecutive pulls may move the estimator away from the mean value of the stationary process.

Another way of understanding the difference between the i.i.d. case and the restless mixing case is to look at the sizes of the confidence intervals around the true value of an arm when the time $t$ to the next pull increases. Given Corollary 1, `Improved-UCB` run in the restless mixing scenario would advocate a pulling strategy based on the lengths $\kappa_k$ of the confidence intervals given by

$$\forall k, \ \kappa_k(t) \doteq |\boldsymbol{\tau}_k|^{-1/2}\sqrt{2(\Lambda_k^{\boldsymbol{\tau}_k} + 2\varphi_k(t - \tau(1)))^2 \log(t)} \tag{8}$$

where $t$ is the overall time index. This shows that working in the i.i.d. case or in the mixing case can imply two different behaviors for the lengths of the confidence interval: in the i.i.d. scenario, $\kappa_k$ has the same form as the classical UCB term (as $\varphi_k = 0$ and $\Lambda_k^{\boldsymbol{\tau}_k} = 1$) and is an increasing function of $t$ while in the $\varphi$-mixing scenario the behavior may be non-monotonic with a decreasing confidence interval up to some point after which the confidence interval becomes increasingly larger. As the purpose of exploration is to tighten the confidence interval as much as possible, the mixing framework points to carefully designed strategies. For instance, when an arm is slow, it is beneficial to wait between two successive pulls of this arm.

By alternating the pulls of the different arms, it is possible to wait up to $K$ unit of time between two consecutive pulls of the same arm. However, it is not sufficient to recover enough independence between the two observed values. For instance, in the case described in (7), after a sequence $\boldsymbol{\tau} = (t_1, t_1 + K, \ldots, t_1 + tK)$, simple calculations give that $\Lambda_k^{\boldsymbol{\tau}} = \mathcal{O}((Kt)^{3/4})$ and the concentration inequality from Corollary 1 for $\widehat{\mu}_k^{\boldsymbol{\tau}}$ reads as $\mathbb{P}(|\mu_k - \widehat{\mu}_k^{\boldsymbol{\tau}}| > \varepsilon) \leq 2 \exp\left(-CK^{3/2}\varepsilon^2 t^{-1/2}\right)$ which entails the same problem.

The problem exhibited above is that if the decrease of the $\varphi_k$ is too slow, pulling an arm in the traditional way, with consecutive pulls, and updating the value of the empirical estimator may lower the certainty with which the estimation of the expected gain is performed. To solve this problem and reduce the confidence interval that are computed for each arm, a better independence between

---

**Algorithm 1** `Remix-UCB`, with parameter $K$, $(\alpha_i)_{i=1\cdots K}$, $T$, $G$ defined in (11)

---

$B_0 \leftarrow \{1, \cdots, K\}, \alpha \leftarrow 1 \wedge \min_{i \in B_0} \alpha_i, \widehat{\mu}^i \leftarrow 0, \ n_0^i \leftarrow 0, \ , k = 1, \dots, K \ , i^* \leftarrow 1$
**for** $s = 1, \dots, \lfloor G^{-1}(T) \rfloor$ **do**
    **Select arm :** If $|B_s| > 1$, then until total time $T_s = \lceil G(s) \rceil$ pull each arm $i \in B_s$ at time $\tau_i(\cdot)$
    defined in (10). If no arm is ready to be pulled, pull the waiting arm $i^*$ instead.
    **Update** :
       1.    Update the empirical mean $\widehat{\mu}^i$ and the number of pulls $n_i$ for each arm $i \in B_s$.
       2.    Obtain $B_{s+1}$ by eliminating from $B_s$ each arm $i$ such that

$$\widehat{\mu}^i + \sqrt{2\frac{(1 + 2\sum_{j=1}^{n_i} \varphi_i(\tau_i(j)))^2 \log(T 2^{-2s})}{n_i}} < \max_{k \in B_s} \widehat{\mu}^k - \sqrt{2\frac{(1 + 2\sum_{j=1}^{n_k} \varphi_k(\tau_k(j)))^2 \log(T 2^{-2s})}{n_k}}$$

       3.    update

$$\alpha \leftarrow 1 \wedge \min_{i \in B_{s+1}} \alpha_i, \quad \text{and} \quad i^* \leftarrow \operatorname*{argmax}_{i \in B_{s+1}} \widehat{\mu}^i + \sqrt{2\frac{(1 + 2\sum_{j=1}^{n_i} \varphi_i(\tau_i(j)))^2 \log(T 2^{-2s})}{n_i}}$$

**end for**

---

the values observed from a given arm is required. This can only be achieved by waiting for the time to pass by. Since an arm must be pulled at each time $t$, simulating the time passing by may be implemented by the idea to pull an arm but not to update the empirical mean $\widehat{\mu}_k$ of this arm with the observed reward. At the same time, it is important to note that even if we do not update the empirical mean of the arm, the resort to the waiting arm may impact the regret. It is therefore crucial to ensure that we pull the best possible arm to limit the resulting regret, whence the arm with the best optimistic value, being used as the waiting arm. Note that this arm may change over time. For the rest of the paper, $\tau$ will only refer to significant pulls of an arm, that is, pulls that lead to an update of the empirical value of the arm.

### 3.2 Algorithm and Regret bound

We may now introduce `Remix-UCB`, depicted in Algorithm 1. As `Improved-UCB`, `Remix-UCB` works in epochs and eliminates, at each epoch, the significantly suboptimal arms.

**High-Level View.** Let $(\theta_s)_{s \in \mathbb{N}}$ be a decreasing sequence of $\mathbb{R}_+^*$ and $(\delta_s)_{s \in \mathbb{N}} \in \mathbb{R}_+^{\mathbb{N}}$. The main idea promoted by `Remix-UCB` is to divide the time available in epochs $1, \dots, s_{max}$ (the outer loop of the algorithm), such that at the end of each epoch $s$, for all the remaining arms $k$ the following holds, $\mathbb{P}(\widehat{\mu}_k^{\tau_k} \geq \mu_k + \theta_s) \vee \mathbb{P}(\widehat{\mu}_k^{\tau_k} \leq \mu_k - \theta_s) \leq \delta_s$, where $\tau_k$ identifies the time instants up to current time $t$ when arm $k$ was selected. Using (4), this means that, for all $k$, with high probability:

$$|\widehat{\mu}_k^{\tau_k} - \mu_k| \leq n_k^{-1/2}\sqrt{2(\Lambda_k^\tau)^2 \log(\delta_s)}. \tag{9}$$

Thus, at the end of epoch $s$ we have, with high probability, a uniform control of the uncertainty with which the empirical rewards $\widehat{\mu}_k^{\tau_k}$ approximate their corresponding rewards $\mu_k$. Based on this, the algorithm eliminates the arms that appear significantly suboptimal (step 2 of the update of `Remix-UCB`). Just as in `Improved-UCB`, the process is re-iterated with parameters $\delta_s$ and $\theta_s$ adjusted as $\delta_s = 1/(T\theta_s^2)$ and $\theta_s = 1/2^s$, where $T$ is the time budget; the modifications of the $\delta_s$ and $\theta_s$ values makes it possible to gain additional information, through new pulls, on the quality of the remaining arms, so arms associated with close-by rewards can be distinguished by the algorithm.

**Policy for pulling arms at epoch** $s$**.** The objective of the policy is to obtain a uniform control of the uncertainty/confidence intervals (9) of all the remaining arms. For some arm $k$ and fixed time budget $T$, such a policy could be obtained as the solution of $\min_{\eta_s, (t_i)_{i=1}^{\eta_s}} t_{\eta_s}$ such that $\frac{(\Lambda^{\tau_s})^2}{n_{s-1} + \eta_s} < \varepsilon$ where the times of pulls $t_i$'s must be increasing and greater than $t_0$ the last element of $\tau_{s-1}$, $\tau_s = \tau_{s-1} \cup (t_1, \dots t_{\eta_s})$ and $n_{s-1}$ (the number of times this arm has already been pulled), $\varepsilon$, $\tau_{s-1}$ are given. This conveys our aim to obtain as fast and efficiently the targetted confidence interval. However, this problem does not have a closed-form solution and, even if it could be solved efficiently, we are more interested in assessing whether it is possible to devise relevant sequences of timepoints that induce a controlled regret, even if they do not solve the optimization problem. To this end, we only focus on

the best sampling rate of the arms, which is an approximation of the previous minimization problem: for each $k$, we search for sampling schemes of the form $\tau_k(n) = t_n = \mathcal{O}(n^\beta)$ for $\beta \geq 1$. For the case where the $\varphi_k$ are not summable ( $\alpha_k \leq 1$), we have the following result.

**Proposition 6.** *Let $\alpha_k \in (0; 1]$ (recall that $\varphi_k(n) = n^{-\alpha_k}$). The optimal sampling rate $\tau_k$ for arm $k$ is $\tau_k(n) = \widetilde{\Theta}(n^{1/\alpha_k})$.*

*Proof.* The idea of the proof is that if the sampling is too frequent (i.e. $\beta$ close to 1), then the dependency between the values of the arm reduces the information obtained by taking the average. In other words, $\sum_n \varphi_k(\tau_k(n))$ increases too quickly. On the other hand, if the sampling is too scarce (i.e. $\beta$ is very large), the information obtained at each pull is important, but the total amount of pulls in a given time $T$ is approximately $T^{1/\beta}$ and thus is too low. The optimal solution to this trade-off is to take $\beta = 1/\alpha$, which directly comes from the fact that this is the point where $\sum_n \varphi_k(\tau_k(n))$ becomes logarithmic. The complete proof is available in the supplementary material. $\square$

If $\alpha_k < 1$, for all $k$, this result means that the best policy (with a sampling scheme of the form $\mathcal{O}(n^\beta)$) should update the empirical means associated with each arm $k$ at a rate $\mathcal{O}(n^{1/\alpha_k})$; contrary to the i.i.d case it is therefore not relevant to try and update the empirical rewards at each time step. There henceforth must be gaps between updates of the means: this is precisely the role of the waiting arm to make this gaps possible. As seen in the depiction of `Remix-UCB`, when pulled, the waiting arm provides a reward that will count for the cumulative gains of the agent and help her control her regret, but that will not be used to update any empirical mean.

As for a precise pulling strategy to implement given Proposition 6, it must be understood that it is the slowest arm that determines the best uniform control possible, since it is the one which will be selected the least number of times: it is unnecessary to pull the fastest arms more often than the slowest arm. Therefore, if $i_1, \ldots, i_{k_s}$ are the $k_s$ remaining arms at epoch $s$, and $\alpha \doteq 1 \wedge \min_{i \in \{i_1, \ldots, i_{k_s}\}} \alpha_i$[1], then an arm selection strategy based on the rate of the slowest arm suggests to pull arm $i_m$ and update $\hat{\mu}_{i_m}^{\tau_{i_m}}$ for the $n$-th time at time instants

$$\begin{cases} (\tau_{i_1}(n-1) + k_s) \vee \lceil n^{1/\alpha} \rceil & \text{if } m = 1 \\ \tau_{i_1}(n) + m - 1 & \text{otherwise} \end{cases} \tag{10}$$

(i.e. all arms are pulled at the same $\mathcal{O}(n^{1/\alpha})$ frequency) and to pull the waiting arm while waiting.

**Time budget per epoch.** In the `Remix-UCB` algorithm, the function $G$ defines the size of the rounds. The definition of $G$ is rather technical: we have $G(s) = \max_{k \in B_s} G_k(s)$ where

$$G_k(s) \doteq \inf \left\{ t \in \mathbb{N}^+, 2(\Lambda_k^\tau)^2 \log(1/\delta_s) \leq t\theta_s \right\} \tag{11}$$

where the $\tau_k(n)$ are defined above. In other words, $G_k$ encodes the minimum amount of time necessary to reach the aimed length of confidence interval by following the aforementioned policy. But the most interesting property of $G$ is that $G(s) = \widetilde{\Theta}((\theta_s^{-2} \log(\delta_s))^{1/\alpha})$. This is the key element which will be used in the proof of the regret bound which can be found in Theorem 2 below.

**Putting it all together.** At epoch $s$, the `Remix-UCB` algorithm starts by selecting the best empirical arm and flags it as the waiting arm. It then determines the speed $\alpha$ of the slowest arm, after which it computes a time budget $T_s = G(s)$. Then, until this time horizon is reached, it pulls arms following the policy described above. Finally, after the time budget is reached, the algorithm eliminates the arms whose empirical mean is significantly lower than the best available empirical mean.

Note that when all the $\varphi_k$ are summable, we have $\alpha = 1$, and thus the algorithm never pulls the waiting arm: `Remix-UCB` mainly differs from `Improved-UCB` by its strategy of alternate pulls. The result below provides an upper bound for the regret of the `Remix-UCB` algorithm:

**Theorem 2.** *For all arm $k$, let $1 \geq \alpha_k > 0$ and $\varphi_k(n) = n^{-\alpha_k}$. Let $\alpha = \min_{k \in \{1, \cdots, K\}} \alpha_k$ and $\Delta_* = \min_{k \in \{1, \cdots, K\}} \{\Delta_k > 0\}$. If $\alpha \leq 1$, the regret of `Remix-UCB` is bounded in order by*

$$\widetilde{\Theta}\left(\Delta_*^{(\alpha-2)/\alpha} \log(T)^{1/\alpha}\right). \tag{12}$$

*Proof.* The proof follows the same line as the proof of the upper bound of the regret of the `Improved-UCB` algorithm. The important modification is the sizes of the blocks, which depend in the mixing case of the $\varphi$ mixing coefficient, and might grow arbitrary large, and the waiting arm, which does not exist in the i.i.d. setting. The dominant term in the regret mentioned in Theorem 2 is related to the pulls of the waiting arm. Indeed, the waiting arm is pulled with an always increasing frequency, but the quality of the waiting arm tends to increase over time, as the arms with the smallest values are eliminated. The complete proof is available in the supplementary material. $\square$

## 4    Discussion and Particular Cases

We here discuss Theorem 2, and some of its variations for special cases of $\varphi$-mixing processes.

First, in the i.i.d case, the regret of `Improved-UCB` is upper bounded by $\mathcal{O}\left(\Delta_*^{-1}\log(T)\right)$ [2]. Observe that (12) comes down to this bound when $\alpha$ tends to 1. Also, note that it is an upper bound of the regret in the algebraically mixing case. It reflects the fact that in this particular case, it is possible to ignore the dependency of the mixing process. It also implies that, even if $\alpha < 1$, i.e. even if the dependency cannot be ignored, by properly using the $\varphi$ mixing property of the different stationary processes, it is possible to obtain an upper bound of polynomial logarithmic order.

Another question is to see what happens when $\alpha_k = 1$, which is an important threshold in our study. Indeed, if $\alpha_k = 1$ the $\varphi_k$ are not summable, but from Proposition 6 we have that $\tau_k(n) \approx O(n)$, i.e. the arms should be sampled as often as possible. Theorem 2 states that the regret is upper bounded in this case by $\widetilde{\Theta}(\Delta_*^{-1}\log T)$. However, it is not possible to know if this bound is comparable to that of the i.i.d case due to the $\widetilde{\Theta}$. Still, from the proof of Theorem 2 we get the following result:

**Corollary 2.** *For all arm k, let $1 \geq \alpha_k > 0$ and $\varphi_k(n) = n^{-\alpha_k}$. Let $\alpha = \min_{k \in \{1, \cdots, K\}} \alpha_k$. Then if $\alpha = 1$, the regret for Algorithm 1 is upper bounded in order by*

$$\mathcal{O}\left(\Delta_*^{-1}\mathcal{G}_\alpha(\log(T))\right) \tag{13}$$

*where $\Delta_* = \min_{k \in \{1, \cdots, K\}}\{\Delta_k > 0\}$ and $\mathcal{G}$ is solution of $\mathcal{G}_\alpha^{-1}(x) = x^\alpha/(\log(x))^2$.*

Although we do not have an explicit formula for the regret in the case $\alpha = 1$, it is interesting to note that (13) is strictly negligible with respect to (12) $\forall \alpha < 1$, but strictly dominates $\mathcal{O}\left(\Delta_*^{-1}\log(T)\right)$. This comes from that while in the case $\alpha = 1$ the waiting arm is no longer used, the time budget necessary to complete step $s$ is still higher that in the i.i.d case.

When $\varphi(n)$ decreases at a logarithmic speed ($\varphi(n) \approx 1/\log(n)^\alpha$ for some $\alpha > 0$), it is still possible to apply the same reasoning as the one developed in this paper. But in this case, `Remix-UCB` will only achieve a regret of $\widetilde{\Theta}\left(\exp\left[(T/\Delta_*)^{1/\alpha}\right]\right)$, which is no longer logarithmic in $T$. In other words, if the $\varphi$ mixing coefficients decrease too slowly, the information given by the concentration inequality in Theorem 1 is not sufficient to deduce interesting information about the mean value of the arms. In this case, the successive values of the $\varphi$-mixing processes are too dependent, and the randomness in the sequence of values is almost negligible; an adversarial bandit algorithm such as `Exp4` [4] may give better results than `Remix-UCB`.

## 5    Conclusion

We have studied an extension of the multi-armed bandit problem to the stationary $\varphi$-mixing framework in the restless case, by providing a functional algorithm and an upper bound of the regret in a general framework. Future work might include a study of a lower bound for the regret in the mixing process case: our first findings on the issue are that the analysis of the worst-case scenario in the mixing framework bears significant challenges. Another interesting point would be the study of the more difficult case of $\beta$-mixing processes. A rather different, but very interesting question that we may address in the future is the possibility to exploit a possible structure of the correlation between rewards over time. For instance, in the case wher the correlation of an arm with the close past is much higher than the correlation with the distant past, it might be interesting to see if the analysis done in [16] can be extended to exploit this correlation structure.

**Acknowledgments.** This work is partially supported by the ANR-funded projet GRETA – Greediness: theory and algorithms (ANR-12-BS02-004-01) and the ND project.

## Footnotes

[1] Since $1/\alpha$ encodes the rate of sampling, it cannot be greater than 1.

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
