[Supplementary Material]

# Supplementary material to: Cornering Stationary and Restless Mixing Bandits with `Remix-UCB`

**Julien Audiffren**
CMLA
ENS Cachan, Paris Saclay University
94235 Cachan France
audiffren@cmla.ens-cachan.fr

**Liva Ralaivola**
QARMA, LIF, CNRS
Aix Marseille University
F-13289 Marseille cedex 9, France
liva.ralaivola@lif.univ-mrs.fr

## A  Proof of Proposition 6

PROOF: In the following, we consider a fixed arm $k$, and to lighten the notation, we drop the explicit dependency on $k$.

Let $T$ be the time budget. Let $\delta$ denote the probability for the concentration inequality to hold.

By assumption, we have $\varphi(n) = 1/n^{\alpha}$ for some $\alpha \leq 1$

The objective is to find the best parameter $\beta \geq 1$, such that pulling the arm at a rate $\tau_{\beta}(n) = \mathcal{O}(n^{\beta})$, given the probability $\delta$ and the time budget $T$, leads to the tightest possible concentration inequality. As specified previously, we only look at the orders of magnitude of this quantity with respect to $T$.

(The case $\beta < 1$ is of no interest because it would require to pull the arm several times at once, which is impossible in our current model.)

Note that with this rate of sampling, the arm is pulled $\mathbf{n} = \tau_{\beta}^{-1}(T) = \mathcal{O}(T^{1/\beta})$ times. The problem we have to handle can be formulated as follows:

$$\arg\min_{\beta \geq 1} \sqrt{\frac{\log(\delta)(1 + \sum_{i=1}^{\mathbf{n}} \varphi(\tau(i)))^2}{\tau_{\beta}^{-1}(T)}}$$

which is equivalent to

$$\arg\max_{\beta \geq 1} \frac{\mathbf{n}}{\log(\delta)(1 + \int_{i=1}^{\mathbf{n}} \varphi(\tau(i)))^2}.$$

We now have three cases to consider:

- $\beta\alpha < 1$

$$\frac{\mathbf{n}}{\log(\delta)(1 + \sum_{i=1}^{\mathbf{n}} \varphi(\tau(i)))^2}$$
$$= \mathcal{O}\left(\frac{\mathbf{n}}{\log(\delta)(\int_{i=0}^{\mathbf{n}} \varphi(\tau(i)))^2}\right)$$
$$= \mathcal{O}\left(\frac{T^{1/\beta}}{\log(\delta)\mathbf{n}^{-2\beta\alpha+2}}\right)$$
$$= \mathcal{O}\left(\frac{T^{1/\beta}}{\log(\delta)T^{-2\alpha+2/\beta}}\right)$$
$$= \mathcal{O}\left(T^{2\alpha-1/\beta}/\log(\delta)\right) = \mathcal{O}\left(T^{\alpha-\varepsilon}/\log(\delta)\right),$$

where in the last line, we used that $\alpha = 1/\beta - \varepsilon$ for some $\varepsilon > 0$.

- $\forall \beta \alpha > 1$

$$\frac{\mathbf{n}}{\log(\delta)(1 + \sum_{i=1}^{\mathbf{n}} \varphi(\tau(i)))^2}$$
$$= \mathcal{O}\left(\mathbf{n}/\log(\delta)\right)$$
$$= \mathcal{O}\left(T^{1/\beta}/\log(\delta) = T^{\alpha - \varepsilon}/\log(\delta)\right),$$

where we successively used that the $1/n^{\alpha\beta}$ are summable, and $\alpha = 1/\beta + \varepsilon$ for some $\varepsilon > 0$.

- $\beta = 1/\alpha$

$$\frac{\mathbf{n}}{\log(\delta)(1 + \sum_{i=1}^{\mathbf{n}} \varphi(\tau(i)))^2}$$
$$= \mathcal{O}\left(\frac{\mathbf{n}}{\log(\delta)(\int_{i=0}^{\mathbf{n}} \varphi(\tau(i)))^2}\right)$$
$$= \mathcal{O}\left(\frac{T^{1/\beta}}{\log(\delta)\log(\mathbf{n})^2}\right)$$
$$= \mathcal{O}\left(\frac{T^{1/\beta}}{\log(\delta)\log(T)^2}\right) = \mathcal{O}\left(\frac{T^{\alpha}}{\log(\delta)\log(T)^2}\right)$$
$$= \widetilde{\Theta}\left(\frac{T^{\alpha}}{\log(\delta)}\right)$$

The optimal result is thus obtained for $\beta = 1/\alpha$. This ends the proof of Proposition 6. $\qquad\square$

# B  Proof of Theorem 2

Let us denote by

$$\Omega_k(n, \delta, \tau) = \sqrt{2\frac{(1 + 2\sum_{j=1}^{n} \varphi_k(\tau(j)))^2 \log(1/\delta)}{n}}$$

the uncertainty of the value of arm $k$ with probability $1/\delta$ after $n$ pulls at rate $\tau(\cdot)$.

The following Lemma gives the length of the rounds used in `Remix-UCB`.

**Lemma B.1.** *Let* $1 < \delta < 0$ *and* $\theta > 0$. *Suppose that* $\varphi_k(n) = \mathcal{O}(1/n^{\alpha_k})$ *and* $\tau_k = \mathcal{O}(n^{1/\alpha_k})$ *with* $0 < \alpha_k \leq 1$. *Then* $T_{k,\delta,\theta}$ *defined as*

$$T_{k,\delta,\theta} = \arg\min\{t \in \mathbb{N}^*, \Omega_k(\tau_k^{-1}(t), \delta, \tau_k) < \theta\}$$

*satisfies*

$$T_{k,\delta,\theta} = \mathcal{O}(\mathcal{G}\left[\theta^{-2}\log(\delta)\right]^{1/\alpha_k})$$

*where* $\mathcal{G}$ *is defined by* $\mathcal{G}^{-1}(x) = x^{\alpha}/(\log(x))^2$.

PROOF: The proof is immediate by definition of $\Omega_k$, and the choice of $\tau_k$, since we have

$$\sqrt{\frac{2(1 + 2\sum_{i=0}^{T_{k,\delta,\theta}^{\alpha}} \varphi(\tau(i))^2 \log(1/\delta))}{T_{k,\delta,\theta}^{\alpha}}} \leq \theta$$

i.e.

$$\frac{2\log(T_{k,\delta,\theta})^2 \log(\delta)}{T_{k,\delta,\theta}^{\alpha}} = \mathcal{O}(\theta^2) \qquad \text{or} \qquad \frac{T_{k,\delta,\theta}^{\alpha}}{\log(T_{k,\delta,\theta})^2} = \mathcal{O}(\theta^{-2}\log(\delta))$$

hence the conclusion. $\qquad\square$

It is important to note that the length of the epoch is almost $\mathcal{O}(\theta^{-2/\beta})$.

We also define, for any arm $k$,

$$m_k = \min\{m \in \mathbb{N}, \quad \Delta_k > \theta_m\},$$

and $m_* = \max_k(m_k)$.

Using Lemma B.1, we can compute the time budget

$$T_s \doteq \max_k T_{k,\delta_s,\theta_s}$$

necessary to obtain, at the end of epoch $s$, that the uncertainty on the values of all the remaining arms is lower than $\theta_s$ with probability $\delta_s$. The consequence of this result for `Remix-UCB` is summarized in the following corollary:

**Corollary B.1.** *At the end of each step $s$ in the* `Remix-UCB` *Algorithm, we have*

  1.
  $$T_s = \widetilde{\Theta}((\theta_s^{-2}\log(\delta_s))^{1/\alpha})$$

  2. $\forall s \geq 0, \forall k$ *a selected arm at step $s$, we have*
  $$\mathbb{P}(\hat{\mu}_k \leq \mu_k + \theta_s) \geq 1 - \delta_s$$

  *and*
  $$\mathbb{P}(\hat{\mu}_k \geq \mu_k - \theta_s) \geq 1 - \delta_s$$

This corollary contains the main ingredients of the proof to upper bound the regret. We use the same proof structure o as the one used in [3]. The Case 1 and 2 follow the same lines as the original proof, and give the same order of regret, but the Case 3 differs greatly due to the waiting arm.

**Case 1 : There exists $k$ such that a suboptimal arm $k$ is not eliminated at step $m_k$, while the optimal arm has not been eliminated**

This happens only if

$$\begin{cases} \hat{\mu}_i \geq \mu_i + \Omega(\tau_k^{-1}(T_{k,\delta,\theta}),\delta_{m_k},\tau_k) \\ \text{or} \\ \hat{\mu}_* \leq \mu_* - \Omega(\tau_*^{-1}(T_{*,\delta,\theta}),\delta_{m_k},\tau_k) \end{cases} \tag{1}$$

The probability of both of these events are controlled by $\delta_m$ thanks to Corollary B.1 and the concentration inequality, thus the probability of this case to occur is lower than

$$\mathcal{O}(\frac{2}{T\theta_{m_i}^2}),$$

and the total regret is upper bounded by

$$\mathcal{O}\left(\sum_i \frac{32}{\Delta_i}\right) = \mathcal{O}\left(\frac{1}{\Delta_*}\right).$$

**Case 2 : The optimal arm is eliminated at step $s$, but during all the previous steps, the suboptimal arms $k$ with $m_k < s$ were correctly eliminated**

The proof in this specific case can be following the same line as in [3], using Corollary B.1 as in Case 1, since the idea is to control the probability of the event to happen. The regret of this case is upper bounded by:

$$\mathcal{O}\left(\frac{1}{\Delta_*}\right).$$

**Case 3 : At step $s$, the optimal arm is still present, and during all the previous steps, the suboptimal arms $k$ with $m_k < s$ were correctly eliminated**

This step is different from the regular `Improved-UCB` analysis. There are indeed two sources of regret to take into account: the regret coming from drawing each of the remaining suboptimal arm $k$ in the evaluation phase, and the regret incurred by pulling the "waiting" arm.

From Corollary B.1, we know that the time budget necessary to complete round $s$ is

$$T_s = \widetilde{\Theta}((\theta_s^{-2} \log(\delta_s))^{1/\alpha})$$

Hence, we can estimate the number of times $\mathcal{N}_s$ a non-waiting arm is pulled up to step $s$,

$$\mathcal{N}_s = \widetilde{\Theta}(\theta_s^{-2} \log(\delta_s)) \tag{2}$$

With this, it is easy to obtain that the first part of the regret in this case is upper bounded by

$$\widetilde{\Theta}\left(\frac{1}{\Delta_*^2}(\log(T))\right)$$

For the second part of the regret, it is important to note that the waiting arm is drawn $\mathcal{O}(T - \min_\alpha T^{\alpha \wedge 1})$ times. Then, either $\forall i\ \alpha_i \geq 1$, and thus the waiting arm is never pulled, leading to no additional regret, or $\exists i$ such that $\alpha_i < 1$.

In the latter case, at each step $s$, the waiting arm $k_s$ is pulled $T_s$ times, but we also know that $m_{k_s} > s$, thus pulling the waiting arm does not generate more than $2\theta_s$ regret.

So the regret coming from the waiting arm at step $s$ is upper bounded (in order of magnitude) by:

$$T_s \theta_s = 2^{-s}\widetilde{\Theta}(2^{2s/\alpha} \log(T2^{-2s})^{1/\alpha})$$
$$= \widetilde{\Theta}(2^{(2s-\alpha)/\alpha} \log(T2^{-2s})^{1/\alpha})$$

Note that after step $m_*$, no additional regret is created by the waiting arm since only the optimal arm remains as a candidate. So the total regret for the waiting arm is upper bounded by:

$$\sum_{s=1}^{m_*} T_s \theta_s = \widetilde{\Theta}\left(2^{m_*(2/\alpha-1)} \log(T)^{1/\alpha}\right)$$
$$= \widetilde{\Theta}\left(\exp(-\log_2(\Delta_*) \log(2)(2-\alpha)/\alpha) \log(T)^{1/\alpha}\right)$$
$$= \widetilde{\Theta}\left(\frac{1}{\Delta_*}^{(2-\alpha)/\alpha} \log(T)^{1/\alpha}\right),$$

since $m_* = -\log_2(\Delta_*)$.

Therefore, the final term of the regret is

$$\widetilde{\Theta}\left(\frac{1}{\Delta_*}^{(2-\alpha)/\alpha} \log(T)^{1/\alpha}\right),$$

hence the conclusion. $\square$