[Reviews · NeurIPS 2015]

Submitted by Assigned_Reviewer_1

Summary: The paper studies the problem of restless mixing bandits, more specifically the case when rewards of an arm evolve according to a \phi-mixing process. In this problem the rewards distribution of an arm is not stationary and evolves according to a \phi-mixing process where the correlation between rewards of an arm in two different rounds decrease over time. The paper proposes an algorithm called Remix-UCB (a variant of Improved-UCB for the stationary case). In this algorithm we maintain confidence bounds on rewards of each arm. Then we proceed in epochs and play each arm certain number of times before eliminating clearly suboptimal arm. The key new idea is that at the end of each of these epochs the algorithm decides to play the best arm certain number of rounds and not use information in these rounds to update confidence bounds. The paper shows regret bound of \theta( (\log T)^{1/\alpha} \delta^(\alpha-1)/alpha) with respect to bench mark of single best arm.

Clarity: The paper is written in a reasonably clear manner.

Significance of the problem: Restless bandits are an important class of bandit problems which model many interesting problems in machine learning on online problems. Any good result should be of interest to the community.

Originality: Although the paper studies the classic problem of restless ucb, the specific variant has not been studied in the literature. This variant looks quite natural and should be of interest. The algorithm that the paper proposes is quite natural and borrows heavily on several techniques in the area including using confidence bounds to estimate mean reward and iterated prunning of suboptimal arms. But the paper also introduces a new idea of playing the best known arm without updating confidence bounds in their algorithm which makes sense in practice.

Comments: The focus of the paper is on getting asymptotic regret which scales like a power of \log(T). If we wanted something of \sqrt{T} regret we can just use exp3 algorithm as the benchmark is single best arm.

Main issues: - One of the interesting aspects of restless bandits is the fact that we can define stronger benchmarks which can play different arms in each round. So the fact that the paper competes against single best arm is a bit weak. - While it is true that we tend to ignore constant in regret bounds the paper is a bit too loose in regret analysis. It does not show dependence on parameters such as k.

- technical correction \lambda_k^\tau = O(t^{3/4}) (line 236 and similarly line 262) - Improved UCB gives interesting bounds when \sum \phi < O(\sqrt{t}) and not just as \sum \phi < constant as the paper indicates.
Summary: The paper studies the problem of restless mixing bandits, more specifically the case when rewards of an arm evolve according to a \phi-mixing process. The paper proposes a new algorithm called Remix-UCB and shows regret of \theta( (\log T)^{1/\alpha} \delta^(\alpha-1)/alpha)

when the mixing coefficient is \phi = O(n^{-\alpha}).

Submitted by Assigned_Reviewer_2

Summary:

The paper proposes a new bandit algorithm call "Remix-UCB".

Remix-UCB looks at the bandit problem where the rewards for each arm are non-iid.

The motivation is to have a solution in situations where rewards in the near future are dependent but in the distance future they are not. For example once you click on an ad now is very unlikely to click on it again in the new future.

This motivation is captured formally by the theory of "mixing of stochastic processes", which basically says that two realizations of a variable are independent when given enough amount of time in between.

The corner stone for developing bound for the remix-UCB is a known concentration inequality for stationary \phi mixing processes.

In this concentration inequality confidence interval are larger in near future but get smaller in distant future.

In essence the bounds behave more similar to standard UCB algorithm when time between arm pulls increases.

But in Bandit algorithms the norm is to decrease the confidence intervals with every pull, which is what happens in the iid case.

The trick of the remix-UCB is to ignore consecutive arm pulls of the same arm when they happen too quickly.

Comments: I found

section 2.1 to be quite compact.

More explanation would be nicer to explain the concentration inequality for mixing processes

Sometimes equations are stated with out clear derivations.

For example equation 8.

The paper has no experimental results:

Overall the paper is not well written and self-contained.

Many of the algorithm parameters are not explained.

For example why is\ theta_s

halved

at each epoch? In equation 9?

What is t_{\eta_s}?

Typos: Equation 2: t_1 is not defined? Line 209: we will "not " recall Line 430: "wher"

Summary: The paper proposes a new UCB algorithm that handles arm pulls that are non-iid.

The assumption is that each arm follows a mixing process.

The authors prove upper bounds for the algorithm.

The paper has two weaknesses.

First, the presentation is not always clear.

Much of the notation is not well defined and paper is quite compact.

Second, there are no empirical results which also leads to doubts about the significance of the work and potential applicability.

Author Feedback
Author rebuttal: We thank the reviewers for their insightful feedback. We here address their main concerns/questions.

On the lack of experiments:
Due to space constraints, we purposely made the choice to focus on providing the reader with all the theoretical elements necessary to understand phi-mixing processes over results from numerical simulations. We made this choice because
we think that it is more important for the paper to give all the tools necessary to understand the concepts the Remix-UCB algorithm is based upon, rather than providing a comparison with other algorithms. This is reinforced by the fact that no other
bandit algorithm has been designed to handle mixing processes.

In any case, we already conducted a set of preliminary experiments to assess the behavior of
our approach in a practical setting. One of the findings is that, for slow mixing processes, we observed that standard best arm algorithms, such as Improved UCB, have a significant chance to erroneously eliminate the
best arm due to heavy time dependency, and thus to give rise to a linear regret. This, in turn, implied the average regrets over several simulations of these classical best-arm bandit identification methods to be way
larger than the average regrets of Remix-UCB.
We intend to include such results (and those obtained from a more comprehensive empirical evaluation) in an extended version of the present paper.

About the "compactness" of the paper:
We agree that the paper is a bit compact, but as stated in the previous section, we already made the difficult choice of removing the experimental section to be able to provide all the necessary elements of theory and algorithmic needed for Remix-UCB.
Since submission time we have devised some changes that will make the reading smoother, though, and we have spotted a few improvements to give lighter and more digest definitions: all this will be included in the final version.

For Reviewer 5:
As we discuss in the paper, Markov processes are a particular case of mixing processes where the dependency
is bounded in time (cf. the diameter of the Markov chain), while the main difficulty addressed
in our paper is the case where the dependency is unbounded and where the sum of the dependencies actually
diverges. This is a far more general setting, and as far as we know, it has never been addressed in the
multi-armed bandit framework.
(On a side note, we would like to point out that the term "iid mixing bandits" used in the review is probably
inaccurate, as Phi-mixing processes are not iid by definition -- unless the Phi-mixing coefficients are all 0.)